# Wireless Sensing Technology Combined with Facial Expression to Realize Multimodal Emotion Recognition

**DOI:** 10.3390/s23010338

**Published:** 2022-12-28

**Authors:** Xiaochao Dang, Zetong Chen, Zhanjun Hao, Macidan Ga, Xinyu Han, Xiaotong Zhang, Jie Yang

**Affiliations:** College of Computer Science & Engineering, Northwest Normal University, Lanzhou 730070, China

**Keywords:** emotion recognition, convolutional neural network, gated recurrent unit, frequency modulated continuous wave, multimodal

## Abstract

Emotions significantly impact human physical and mental health, and, therefore, emotion recognition has been a popular research area in neuroscience, psychology, and medicine. In this paper, we preprocess the raw signals acquired by millimeter-wave radar to obtain high-quality heartbeat and respiration signals. Then, we propose a deep learning model incorporating a convolutional neural network and gated recurrent unit neural network in combination with human face expression images. The model achieves a recognition accuracy of 84.5% in person-dependent experiments and 74.25% in person-independent experiments. The experiments show that it outperforms a single deep learning model compared to traditional machine learning algorithms.

## 1. Introduction

Emotions significantly impact people’s health that cannot be ignored. Studies have shown that malfunctions in mood and emotion regulation are the cause of many mental illnesses [1], and people who maintain negative moods for long periods not only have low immunity [2] but also perform worse in terms of memory [3]. Especially in recent years, traditional lifestyles have undergone a dramatic shift influenced by COVID-19, and online learning and working have become a common way of being adopted. The lack of communication and outdoor activities for a long time has affected people’s mood and emotion regulation to a greater extent. Therefore, emotion recognition in human-computer interaction has become a task that cannot be ignored.

Wireless sensing technology realizes the target’s sensing task by analyzing the target’s influence on the surrounding wireless signals. Compared to traditional sensing techniques, its features of not requiring the target to wear any sensor, non-line of sight propagation, and privacy protection have made it an ideal solution for sensing tasks in many fields, such as intrusion detection [4], gesture recognition [5], physiological detection [6] and indoor positioning [7]. As wireless sensing tasks expand from coarse-grained to fine-grained, traditional technologies such as ZigBee, CSI, and CW signals can no longer meet the current sensing needs, and millimetre-wave radar has received widespread attention due to its larger bandwidth, more sensitive distance resolution, and higher Doppler frequency. Millimeter wave radar has achieved remarkable results in theoretical research [8] and practical application of fine-grained perception [9,10], which also provides ideas for us to realize the task of contactless emotion recognition.

The study of wireless sensing technology to achieve emotion recognition involves several disciplines, such as neurology, psychology, and computer science. Emotion theory is the basis for wireless sensing technology to achieve emotion recognition, which is an extremely complex mental state [11]. Many emotion theories have been proposed in psychology to explain human emotions. The models of emotion theories used in past studies are divided into two main categories: one is the basic emotions applicable to different cultures and ages, the most classic being the six basic emotions proposed by Ekman in the 1970s: happy, sad, surprise, fear, anger, and disgust [8]. Another class of emotion models that quantify emotions, such as the three-dimensional emotion classification system designed and proposed by Wilhelm Wundt, is the ring model [12]. The three axes describe the potency, arousal, and intensity of emotions. Moreover, the more used in many past studies is a two-dimensional model based on the ring model, where validity describes the range of emotions from negative to positive, and arousal describes the range of emotions from positive to negative. In this paper, we use Russell’s validity-arousal two-dimensional emotion model, which can quantify all emotions in a two-dimensional emotion system [13,14].

Current approaches to emotion recognition fall into two broad categories. The first category is traditional machine learning (ML) methods, which are based on using a specific model and require elaborate and manual extraction of relevant features as input to the classification model. The second category is the more commonly used deep learning (DL) methods, which do not have a specific model and can automatically learn the internal principles of the collected data and automatically complete the extraction of features [15,16]. Moreover, the reasons for the rise of deep learning methods are multifaceted. First, as mentioned earlier, since the extracted features are crucial for the classification performance of ML models, manually extracting features is time-consuming and laborious. Second, the features obtained with difficulty usually only solve problems in a particular domain and are difficult to reuse.

This paper’s proposed overall emotion recognition framework is shown in Figure 1. Millimeter wave radar is used to monitor the subject’s physiological information while a computer camera records the facial expressions. Subsequently, the Moving Target Inidcation (MTI) method is used in the signal pre-processing stage to eliminate the background noise and the effect caused by the measurement instrument. Then the Variational Modal Decomposition (VMD) method extracts and separates the signals to obtain the heartbeat and respiration signals. Furthermore, for the obtained signals, this paper is designed using 1D-CNN, 2D-CNN and GRU stacked deep learning models, which have good performance in subject-related and subject-independent experiments, and later the method of this paper is referred to as ER-MiCG. The contributions of this paper are as follows:In this paper, we use millimeter-wave radar to capture heartbeat and respiration signals in different emotional states while combining three modal data of facial expression images, and then perform parallel fusion after feature extraction by respective CNN deep learning models and use the fused features as the input of GRU deep learning model, and then achieve the classification task of four emotions.Considering that the breathing and heartbeat signals are non-stationary signals and the noise problem of the experiment scene, the method of combining MTI and VMD is proposed, and the comparison experiment with an MI5 smartwatch proves that it can get high-quality signals.To prove the effectiveness and superiority of the proposed method, this paper not only sets up comparison tests with traditional machine learning methods and single deep learning models but also a comprehensive comparison from robustness and other advanced methods. The model is proven to have a more excellent classification ability through many experiments.

## 2. Relate Work

In recent years, considering the great influence of emotions on people’s physical and mental health, the work on emotion recognition has a broad application prospect, which is one of the reasons why this field has received close attention from the industry [17,18,19,20,21]. Since emotional fluctuations cause changes in the external expressive reflections of the human body, researchers have extracted emotion-related features through facial expressions [22], body movements [23,24], and voice intonation [25]. It also causes changes in human physiological signals, so researchers often capture a certain physiological signal with the help of unique equipment, such as EEG signals [15,16], ECG signals [20], and skin electrical activity [26]. Usually, the methods using physiological signals perform better in robustness and recognition accuracy because physiological signals cannot be artificially hidden or altered.

Research related to emotion recognition consists of the following steps: emotion stimulation, data acquisition, data preprocessing, feature extraction, and finally, recognition and classification. In the emotional stimulation phase, pictures, and videos evoke a particular emotion in the subject. The data acquisition phase is even more diverse. In psychology, most use more complex, expensive, and highly invasive specialized equipment, so this paper aims to reduce the deployment cost, system complexity, and invasiveness of the emotion recognition task using wireless sensing technology. The preprocessing phase usually results in the time domain, frequency domain, time-frequency, or spatial domain signals, and feature extraction from a single source or a combination of multiple sources is a common approach [27]. The final stage of identification is usually done by ML or DL methods.

To date, traditional ML methods have proposed many well-performing classification algorithms such as support vector machine (SVM), random forest (RF), linear discriminant analysis (LDA), and Bayesian [28], and these algorithms are being improved as they are continuously studied in depth in various fields. Mansour Sheikhan et al. [29] innovatively proposed a modular neural-SVM classifier that inputs a total of 55 features with MFCCs, velocity, acceleration, and logarithmic energy as basic features and achieves the recognition of three emotional states. Tiwari et al. [30] designed and proposed a novel algorithm of Shifted Delta Acceleration Linear Discriminant Analysis (SDA-LDA) algorithm to extract the most adjudicative and robust features in speech sequences and video sequences. The method is shown to have high recognition accuracy on all four public datasets.

Researchers have gradually focused their attention on DL techniques to address the many challenges encountered in traditional machine learning and its limitations [31]. Convolutional neural networks (CNN), deep confidence networks (DBN), recurrent neural networks, hybrid networks, and other classification models have been proposed successively to handle various classification tasks [32]. Similarly, these models have been improved as the complexity of classification tasks continues to increase and the real-time and recognition accuracy requirements improve. Mustaqeem and Soon-il Kwonwei [33] proposed a one-dimensional dilated convolutional neural network (DCNN) to solve the problem of lack of real-time processing in speech emotion recognition. Velagapudi Sreenivas et al. [34] used not only wrapper-based feature selection techniques to reduce the complexity of the recognition framework but also optimized the weight parameters of the DBN network by the Harris Hawk optimization algorithm and demonstrated that the proposed method outperformed not only existing methods on three datasets but also outperformed existing methods on six emotional states. The proposed method not only proves to be superior to existing methods on three datasets but also has high recognition accuracy in the recognition of six emotional states.

With the popularity of WIFI and extensive research on wireless sensing technologies such as millimeter-wave radar in recent years, research on emotion recognition has taken on a whole new dimension. However, compared to CSI signals emitted by WIFI devices, millimeter-wave radar emits more directional signals, which means that the multipath effect is significantly reduced. Moreover, the short wavelength makes it more sensitive to small movements. There are three types of millimeter wave radars commonly used in current research: continuous wave radar (CW), ultra-wideband radar (UWB), and frequency-modulated continuous wave radar (FMCW). The reason for using FMCW radar in this paper is that CW radar has unsolved problems in acquiring static information about the target location or the environment. In contrast, UWB radar has a small coverage area due to the limitation of transmitting power. The FMCW radar has not only good performance in coarse-grained activity recognition, such as precise indoor positioning [35], dynamic gesture recognition [36], human activity recognition and fall detection [37] but also excellent performance in fine-grained activity recognition, such as human vital signs detection and tracking [38,39].

## 3. The Proposed Method

### 3.1. Working Principle of FMCW Radar

Standard FMCW radar signals are divided into frequency shift keyed continuous wave (FSKCW), stepping frequency continuous wave (SFCW), and linear frequency modulated continuous wave (LFMCW) according to the frequency waveform. However, the MMW radar in this paper transmits signals using linear frequency-modulated continuous waves whose frequency increases linearly within the modulation period. Namely, the waveform is sawtooth. Then the frequency change of the FMCW signal with time within a single modulation cycle can be expressed as
(1)$$f=f_{c}+\frac{B}{T_{c}}t,$$
where $f_{c}$ is the starting frequency of the chirp signal, *B* is the bandwidth of the chirp signal, and $T_{c}$ is the chirp signal duration. The target detection of radar is to obtain the position, velocity and angle of the target by the amplitude, phase, and frequency of the reflected signal. When the human body carries out normal breathing and heartbeat, it will cause changes in different parts of the body’s surface. Because the chest position is close to the lung and heart, the position of the chest changes most obviously. This is also the choice of this article through the displacement of the chest to achieve the measurement of heartbeat and respiration. The millimeter wave transmitted signal and reflected signal can be expressed as
(2)$$x_{T}\left(t\right)=A_{TX}\cos\left(2\pi f_{c}t+\pi \frac{B}{T_{c}}t^{2}+\phi \left(t\right)\right),$$
(3)$$x_{R}\left(t\right)=A_{RX}\cos\left[2\pi f_{c}\left(t-t_{d}\right)+\pi \frac{B}{T_{c}}\left(t-t_{d}\right)+\phi \left(t-t_{d}\right)\right],$$
where ϕ(t) is phase noise, $t_{d}$ is the propagation time delay generated by radar signal propagation between radar and target. At this time, the transmitted signal and the echo signal are mixed, and then the high-frequency signal is filtered by the low-pass filter to obtain the IF signal. Then the IF signal can be expressed as
(4)$$\begin{array}{c} IF\left(t\right)=A_{TX}A_{RX}\exp\left(j\left(2\pi f_{c}t_{d}+\pi \frac{B}{T_{c}}t_{d}t+\pi \frac{B}{T_{c}}{t_{d}}^{2}+\Delta \phi \left(t\right)\right)\right)\\ \approx A_{TX}A_{RX}\exp\left(j\left(2\pi \frac{B}{T_{c}}t_{d}t+2\pi f_{c}t_{d}\right)\right)\\ =A_{TX}A_{RX}\exp\left(j\left(4\pi \frac{BR}{cT_{c}}t+\frac{4\pi R}{\lambda }\right)\right)\\ =A_{TX}A_{RX}\exp\left(j\left(2\pi f_{IF}t+\varphi_{IF}\right)\right), \end{array}$$
where $\pi \frac{B}{T_{c}}{t_{d}}^{2}$ is negligible, λ is the wavelength, *R* is the distance from the radar to the chest wall, $f_{IF}=\frac{2BR}{cT_{c}}$ is the frequency of the IF signal, and $\varphi_{IF}=\frac{4\pi R}{\lambda }$ is the phase of the IF signal. In actual measurement, the chest cavity will be slightly displaced in different degrees along with respiration and heartbeat, so R=Δd+d(t). Therefore, Δd is the relative distance between the radar and the detected target, and d(t) is the displacement of the chest wall over time. Then, the above equation can be rewritten as
(5)$$\begin{array}{c} IF\left(t\right)=A_{TX}A_{RX}\exp\left(j\left(\frac{4\pi (\Delta d+d(t))}{\lambda }+4\pi \frac{B(\Delta d+d(t))}{cT_{c}}+\Delta \phi \left(t\right)\right)\right)\\ =A_{TX}A_{RX}\exp\left(j\left(2\pi f_{IF}t+\varphi_{IF}\right)\right), \end{array}$$
(6)$$f_{IF}=\frac{2B(\Delta d+d(t))}{cT_{c}},$$
(7)$$\varphi_{IF}=\frac{4\pi (\Delta d+d(t))}{\lambda }.$$

For the millimeter wave radar with a center frequency of 77 GHZ used in this paper, chirp signal duration $T_{c}$ is set to 40 μs, and frequency slope *k* is 50 MHZ/μs. When the measured target is displaced by 0.1 mm along the electromagnetic wave direction of the radar, it can be obtained that the frequency change is about 33.3 Hz according to the above formula. Only about 0.001 cycles were changed, making it difficult to recognize the frequency changes caused by human respiration and heartbeat on the spectrum. The displacement of the human chest can also change the phase of the echo signal. If the above assumption remains unchanged; it can be concluded from the above equation that when the measured target is displaced by 0.1 mm along the electromagnetic wave direction of the radar, the phase change of its signal is about 18 degrees. Compared with identifying the frequency changes caused by human respiration and heartbeat in the spectrum, it is easier to identify the phase of the target. Therefore, it is feasible for radar to detect the phase changes of the received echo signal to obtain the respiratory and heartbeat information of the target.

### 3.2. Data Preprocessing

#### 3.2.1. MTI Removes Static Clutter

In the experimental environment of this paper, static objects such as laptops, desks, and chairs will reflect echo signals with reliable energy, sometimes even causing the detection target to be submerged, which severely impacts the detection of physiological signals. Therefore, it is necessary to deal with static clutter. Considering the static object echo signal will not change over time, and the chest in the ups and downs when motion over time leads to the echo signal, therefore, adopt the MTI method to deal with the original signal: *n* take size for the sliding window, for the same distance average continuous n slow sampling point of the unit, as the distance unit in a static environment noise component; Then subtract the mean value from all the slow sampling data in the sliding window. The calculation result is again averaged as the result of removing the static clutter component at the current time. Figure 2 shows the original radar echo signal and the echo signal after MTI processing. It can be seen that the static clutter has been filtered out.

#### 3.2.2. VMD Algorithm

After the static clutter is processed, a two-dimensional fast Fourier transform (2D-FFT) is carried out on the signal to extract the phase value of the target. Then phase unwinding is carried out on the signal. Finally, since the frequency range of breathing and heartbeat is between 0 and 3.3 HZ. Therefore, a fourth-order Butterworth filter with a cutoff frequency of 3.3 HZ was used in this paper to filter high-frequency noise. Subsequently, the VMD algorithm was selected because of its high efficiency and effective suppression of non-stationarity of vital signs signals [40]. Finally, the signals were decomposed into a series of IMF components. VMD algorithm is an adaptive and quasi-orthogonal decomposition method based on cyclic iteration to obtain the optimal solution of the constrained variational problem, determine different frequency centers and bandwidth, and then decompose the original vibration signal into a series of eigenmode functions of different frequencies. Its central core is to construct the variational problem and solve the variational problem, which is solved based on the classical Wiener filter, frequency mixing, and Hilbert transform. In this work, we set the IMF of physiological sign signals as the central frequency of IMF signals and k as the number of IMF components. Then the unconstrained variational problem of physiological signal decomposition is defined as follows [41]: (8)$$\begin{array}{cc} L\left(\left\{u_{k}\right\},\left(w_{k}\right),\lambda \right)= & \alpha \sum_{k=1}^{k}{∥\partial_{t}\left[\left(\delta \left(t\right)+\frac{j}{\pi t}\right)*u_{k}\left(t\right)\right]e^{-jww/t}∥}_{2}^{2}+{∥f\left(t\right)-\sum_{k=1}^{k}u_{k}\left(t\right)∥}_{2}^{2}\\ \; & +\langle \lambda \left(t\right),f\left(t\right)-\sum_{k=1}^{k}u_{k}\left(t\right)\rangle . \end{array}$$

After *n* iterations, and are calculated as follows: (9)$${\hat{u}}_{k}^{n+1}\left(\omega \right)=\frac{\hat{f}\left(\omega \right)-\sum_{i\neq k}{\hat{u}}_{k}\left(\omega \right)+\frac{\hat{\lambda }\left(\omega \right)}{2}}{1+2\alpha {\left(\omega -\omega_{k}\right)}^{2}},$$
(10)$${\hat{\omega }}_{k}^{n+1}\left(\omega \right)=\frac{\int_{0}^{\infty }\omega {\left|{\hat{u}}_{k}\left(w\right)\right|}^{2}d\omega }{\int_{0}^{\infty }{\hat{u}}_{k}{\left|\left(\omega \right)\right|}^{2}d\omega }.$$

On the type of α is the introduction of quadratic penalty factor, λ(t) is the introduced Lagrangian operator, *k* is after VMD decomposition to get the number of intrinsic mode function, δ(t) is Dirac function, $e^{-jwkt}$ is a complex signal in time the rotation of the phasor, and $u_{k}^{n+1}$, u(ω) and f(ω) Fourier transform with ${\hat{u}}_{k}^{n+1}$, $\hat{u}\left(\omega \right)$ and $\hat{f}\left(\omega \right)$, respectively. Using VMD, the signal nonlinearity can be decomposed into IMF sets with specific sparsity. Specifically, VMD divides the original signal acquired by millimeter-wave radar into *k* discrete IMF components, and the value of k must minimize the sum of the bandwidths of all components. Then Hilbert transform is applied to the decomposed modal function to obtain the analytic signal and its unilateral spectrum related to each mode. Then, the analytic signal’s center frequency is estimated, the exponential term is added for adjustment, and the spectrum is modulated to the corresponding fundamental frequency band. Finally, the bandwidth is estimated according to the Gaussian smoothness of the demodulation signal, and the norm of gradient L2 is calculated [41].

When determining the *k* value, when the central frequency of each IMF component remains stable at different *k* values, it indicates that the original physiological data is not over-decomposed. When the frequency ratio of adjacent centers is greater than 90%, it indicates that the original physiological data is over-decomposed, and a faulty component is obtained by over-decomposition in the IMF component. In this paper, all IMF is stable when *k* is not less than 6, which also means that signal decomposition is completed when *k* is not more than 6. In addition, when *k* = 7, the central frequency of adjacent IMF is greater than 90%, the decomposition is excessive, so *k* = 6 is finally determined. A group of results obtained by the VMD algorithm is shown in Figure 3.

Then, the frequency spectrum of each IMF was analyzed, and the average instantaneous frequency was calculated. The respiratory signal could be reconstructed by adding the IMF components, whose average frequency was in the range of [0.1–0.6 HZ]. In contrast, the heartbeat signal could be reconstructed by adding the IMF components, whose average frequency was in the range of [0.8–3.3 HZ]. The reconstructed breathing and heartbeat signals under different emotions are shown in Figure 4 and Figure 5, respectively.

The obtained heartbeat signal is subjected to Fourier transform to obtain the heartbeat frequency. We collected two hundred sets of heartbeat data by allowing the subjects to wear the MI5 smartwatch to participate in the experiment. The heartbeat frequency observation value of the MI5 smartwatch and the heartbeat frequency obtained by the proposed method is used for linear regression modeling. Figure 6 shows the proposed method’s overall fitting degree ($R^{2}$). The closer the score of $R^{2}$ is to 1, the closer the fitted curve is to the actual curve. The score of the proposed method is 0.94, which also shows that the accuracy and reliability of the heartbeat signal obtained in this paper are excellent.

#### 3.2.3. Video Signal Preprocessing

In this paper, the acquisition of physiological signals and facial video data are synchronized, and each video sample has the same time. Therefore, the preprocessing of the video signal starts with selecting key frames from the 60 s video segments. Specifically, first, we use the LBP-AdaBoost algorithm to detect the face in each frame and crop the frame to the largest face that can be displayed. Subsequently, in a window of 2*i* + 1 frames, set *i* to 5 (based on experience) and calculate the histogram of each frame in the window, applying the cardinality distance to find the difference between the histograms of consecutive frames and the one with the smallest difference is selected as the keyframe. The window is then moved by 8 frames to continue selecting keyframes and continues until the end of the video. Finally, the size of the key image frame is adjusted to match the input size of the subsequent recognition model, and the final cropped facial image for each frame is uniformly adjusted to 227×227×3.

### 3.3. Proposed Deep Learning Model

Based on the emotional data of different dimensions obtained by the above method, this paper proposes a deep learning model, as shown in Figure 7. The data obtained are passed through four convolution modules with similar structures. The dimension of the data determines the specific parameters of each module. The data of three channels with different dimensions can obtain richer feature information. After that, the obtained feature elements are spliced in parallel as the input of the GRU neural network. Then the emotion recognition is realized by the dense layer with the softmax activation function as the classification function.

#### 3.3.1. Construction of 1D-CNN

In many emotion recognition-related works, 1D-CNN has been proven to perform well in feature extraction of one-dimensional sequence information [42]. According to the characteristics of respiratory and heartbeat signals extracted in this paper, the proposed 1D-CNN model is shown in Figure 7. It consists of four convolution blocks, each of which consists of a convolution layer, a top pooling layer, a batch normalization layer, and an ELU activation function layer. The batch standardization layer is added to alleviate the internal covariate shift and improve the feature extraction ability. ELU activation function is a new activation function that combines the left soft saturation of the sigmoid activation function and the right non-saturation of the ReLU activation function. The advantage of using this activation function is that the right linear part makes the ELU alleviate the gradient disappearance problem, and the left soft saturation performance makes the ELU more robust to input changes or noise. The specific parameter settings are shown in Table 1.

#### 3.3.2. Construction of 2D-CNN

Facial expression is an external expression of emotion, and many previous studies have shown that facial expression is an important measure in emotion recognition, so it is necessary to combine the features of facial expression to achieve emotion recognition. The 2D-CNN constructed in this paper has the same structure as the 1D-CNN mentioned above, which also consists of four convolutional blocks. However, the input is a face image of size 227×227×3 obtained from video processing. The convolutional layer serves as the core to detect local features of the facial image by learning filter banks. The pooling layer functions to gradually reduce the size of the representation space to reduce the parameters and computation in the network, thus, controlling overfitting, also using the ELU activation function. Each layer’s convolution and pooling kernels are two-dimensional, and the specific parameter settings are shown in Table 2.

#### 3.3.3. Construction of GRU

The use of GRU is for better recurrent neural network hidden layer variable gradient may appear attenuation or explosion problem, but also to better capture the time series data interval more extensive dependencies. Compared with the LSTM network structure, GRU improves the training speed of the model while reducing the training parameters required for its internal network structure training. In this paper, emotion-related features are extracted from one-dimensional sequence information on human physiology and image information of facial expression changes. Then these features of different scales are fused as input of GRU to mark time-related sequences for enhancing feature representation. In addition, GRU makes the whole network model fault-tolerant and can predict and erase the wrong channels of the corresponding feature map at a particular time according to other features, A GRU layer with 64 neurons is set up.

## 4. Experiments and Results

### 4.1. Experimental Design

The Texas Instruments IWR1642 single-chip FMCW millimeter wave radar operating in the 76–81 GHz band was used in the experiment, along with the DCA1000 in streaming data mode to collect raw human heartbeat and respiration data, and Table 3 shows the specific parameter settings of the radar. The experiments were conducted in a relatively confined and quiet environment, during which 15 volunteers (8 males and 7 females) were recruited to participate in the experiments, during which the subjects were required to sit as still as possible in front of the millimeter wave radar to watch a video designated for inducing specific emotions. All the video clips were taken from a previous study [40], which included 60 videos (15 for each emotion) that induced four emotions: happy, angry, relaxed, and sad. In addition, nine videos that put the person in a neutral emotional state were included, each serving as a baseline signal at the onset of inducing a specific emotion, thus, eliminating the daily dependence on physiological data. Experimental labeling continued based on SAM forms that subjects filled out after viewing the videos and labeled the data according to the two-site emotion model proposed by Russell. Nine hundred sets of one-dimensional sequential information on heartbeat and respiration and corresponding facial videos were collected. Of these, 36 sets of data were discarded for reasons such as blurred picture quality and the presence of large movements of the subjects. Of the remaining data, 86 sets were separated and used as the validation set, another 86 sets were separated and used as the test set, and 692 sets were used as the training set.

### 4.2. Evaluating-Indicator and Cross Validation

To evaluate the classification ability of the proposed method, this paper uses four evaluation indexes: accuracy, precision, recall rate, and *F-score*. It can be calculated by Formulas (11)–(14): (11)$$accuracy=\frac{TP+TN}{TP+FP+TN+FN}\times 100\%$$
(12)$$precision=\frac{TP}{TP+FN}\times 100\%$$
(13)$$\mathrm{recall}=\frac{TP}{TP+TN}\times 100\%$$
(14)$$F\text{-}score=\frac{2\times recall\times precision}{recall+precision}$$

Among them, *TP* was true positive, *TN* was true negative, *FP* was false positive, and *FN* was false negative. The recognition rate refers to the ratio of the number of samples correctly classified by the optimal classifier trained by the training data to the total number of samples for the total test data set. Accuracy refers to the ratio of the number of real positive samples classified in the test data set to the number of positive samples classified. Recall rate refers to the ratio of the number of samples classified as true positive examples to the number of samples of all true positive examples. F-score is an extension of the recognition rate, combining precision and recall. Based on the above metrics and considering that the data volume in this paper is not huge, we introduce leave-one-out cross-validation (LOOCV) to validate the predictive ability of the model further. This method involves taking out one data at a time as the unique elements of the test set in a dataset composed of *n* data, while the other *n* − 1 data are used as the training set for training the model and tuning the reference. The result is that we end up training *n* models, each time getting a Mean Squared Error (MSE), and calculating the final test MSE is averaging these *n* MSEs, so the mathematical expression is: (15)$$CV_{\left(n\right)}=\frac{1}{n}\sum_{i=1}^{n}MSE_{i}$$

Since the degree of emotional expression is highly correlated with the level of the video stimuli, and also each person’s emotion expression for the same video stimuli varies, the proposed model’s LOOCV accuracy of 72.83% has reached a high score. In addition, the accuracy, recall, and F1 scores after LOOCV are 0.721, 0.728, and 0.723, respectively, which proves that the model has a strong prediction and generalization ability.

### 4.3. Experimental Analysis

#### 4.3.1. Emotion Recognition Accuracy of ER-MiCG

To evaluate the classification accuracy of ER-MiCG, we trained two types of classifiers using the data of the 15 subjects mentioned above. The first type is a person-related classifier, i.e., each subject is trained separately under a specific topic. The second type is a person-independent classifier, i.e., the same classifier is used for all subjects. Moreover, the results are as expected. The person-dependent classifier has significantly higher recognition accuracy than the person-independent classifier, which is also consistent with the extensive previous research literature. The Figure 8 shows that the average recognition accuracy of the four emotions in the person-related classifier reaches 85.5%. In contrast, the recognition accuracy of the person-independent classifier reaches 74.25%, which means that ER-MiCG is more successful in achieving the emotion recognition task by automatically extracting emotion-related features.

#### 4.3.2. Robustness Testing

The expression of emotion in physiological signals is subjective, reflected in the different fluctuations of physiological signals when everyone is in the same emotional state. At the same time, each person shows different physiological characteristics in the same emotional state on different days. This is due to each person’s different sleep conditions and material intake on the day, so it is necessary to evaluate the method proposed in this paper on different days. We collected the experimental data of six subjects in the experimental environment of this paper for seven days a week. As shown in the Figure 9, the average recognition rate of the four emotions is higher than 72%, which shows that our method is less affected by time factors.

In addition, the distance between the device and the subject is also an essential factor affecting the recognition accuracy because of the short wavelength of the millimeter wave, the characteristics of easy attenuation, and the video image taken by the camera will also decrease as the distance increases. Therefore, we adjusted the distance between the millimeter wave radar and the subjects in the experiment and collected the experimental data of 10 subjects at the distance of 40 cm, 50 cm, 60 cm, and 70 cm, respectively. Figure 10 shows the recognition accuracy of four emotions at different distances. It can be seen that the average recognition accuracy of the four emotions gradually decreases with the increase in distance.

#### 4.3.3. Comparison of Different Emotion Recognition Methods

The idea of our work comes from similar previous studies. In this paper, we have selected studies using wireless signals or facial expression images to achieve emotion recognition in recent years for comparison. Table 4 shows the differences between the four methods and ER-MICG in four dimensions, where the average recognition accuracies are obtained with person-independent classifiers. Among them, facial expression-based methods have higher recognition accuracy because the research started earlier and the technology is very mature. The limitation of such methods is that facial expressions can be deliberately hidden or changed. In contrast, physiological information cannot be deliberately changed. Therefore, combining the two techniques can compensate for the limitations of single-channel data and make the recognition results more convincing. However, the research in this field is in its initial stage. As the research progresses, combining physiological information obtained by wireless sensing technology with facial expressions will achieve higher recognition accuracy. From Table 4, we can see that the recognition accuracy of the proposed method in this paper is higher than that of using only a single wireless signal and other similar methods.

#### 4.3.4. Comparison with Traditional Machine Learning Algorithms

ML methods in previous studies have solved many problems, but there is no denying that their manual feature extraction is time-consuming and laborious. Researchers have turned their attention to DL because the method can overcome the difficulty of obtaining valid time series features from time series data. DL alleviates the burden of extracting manual features for ML models. Instead, it can automatically learn hierarchical feature representations. This eliminates the need for data pre-processing and feature space reconstruction in a standard ML pipeline. In this paper, to further demonstrate the classification ability of the deep learning model proposed in this paper, and traditional machine learning models, statistical features of emotions are extracted using statistical feature recognition methods. Person-related classification models are constructed using support vector machine (SVM) and random forest (RF), commonly used in previous studies. Experimental data from ten subjects are used for comparison in the recognition accuracy dimension. The Figure 11 shows the classification of the three classification models for the four emotions, and MiCG has a higher classification accuracy than SVM and RF for all four emotions.

#### 4.3.5. Comparison of Different Deep Learning Models

In this paper, CNN and GRU deep learning models are used in the emotion recognition task according to the characteristics of different modalities. Then the classification ability of ER-MiCG needs to be validated compared to these two classification models. In addition, the intensity of a specific emotion induced by audiovisual stimuli is highly subjective, which increases the difficulty of completing the classification task only under commonalities. Therefore, it is crucial to validate the classification ability of the proposed classification model ER-CG in this paper. We compare ER-MiCG with CNN and GRU deep learning models. The receiver operating characteristic curve (ROC curve) is commonly used to evaluate the classification ability of the model. The horizontal coordinate *X*-axis is 1 − specificity, also known as the false positive rate (false alarm rate), and the closer the *X*-axis is to zero, the higher the accuracy rate; the vertical coordinate *Y*-axis is called sensitivity, also known as the true positive rate, and the larger the *Y*-axis represents the better the accuracy rate. According to the curve position, the whole graph was divided into two parts. The area of the part under the curve is called AUC (Area Under Curve), which is used to indicate prediction accuracy. The higher the AUC value, the larger the area under the curve and the higher the prediction accuracy. Figure 12 shows that the AUC scores in the ROC curves of the three models are in descending order: ER-MiCG > GRU > CNN, which also indicates that ER-MiCG has a higher classification ability.

## 5. Conclusions

In this study, we used wireless sensing technology to acquire physiological signals from subjects and captured videos of their facial expressions via a camera. The original signals were then removed from static clutter and separated and reconstructed to obtain higher quality time domain signals of heartbeat and respiration, cropping, and keyframe selection of facial expression images. Finally, a deep learning model combining CNN and GRU is designed based on their respective characteristics to realize the recognition of four emotional states. The model combines the powerful feature representation capability of CNN in the face of data with different dimensions and the GRU model’s simplicity, efficiency, and stronger fault tolerance. Extensive experiments have shown that the heartbeat breathing signal extraction and processing method proposed in this paper has good accuracy. In addition, the classification model proposed in this paper performs better than single deep learning models and traditional machine learning models. The future work is to improve the method further. The presence of multiple people, people in large movements or motion, poor lighting conditions, and other complex real-world environments can have a significant impact on the classification results, so we hope that the work done so far can lay the foundation for solving the above limitations and even applying them to clinical routines in the future.

## Figures and Tables

**Figure 1 sensors-23-00338-f001:**
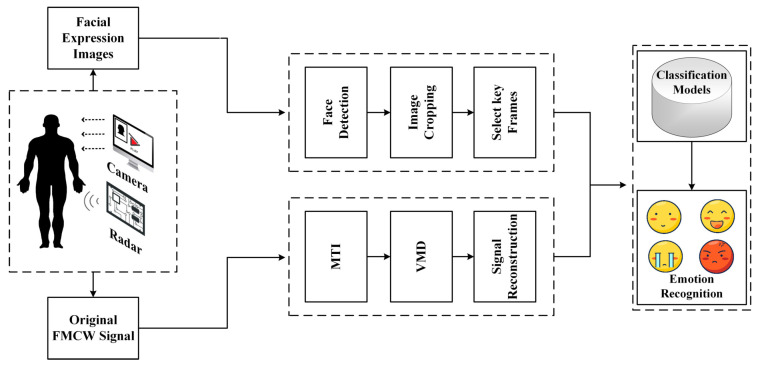
The overall frame diagram of the ER-MiCG method.

**Figure 2 sensors-23-00338-f002:**
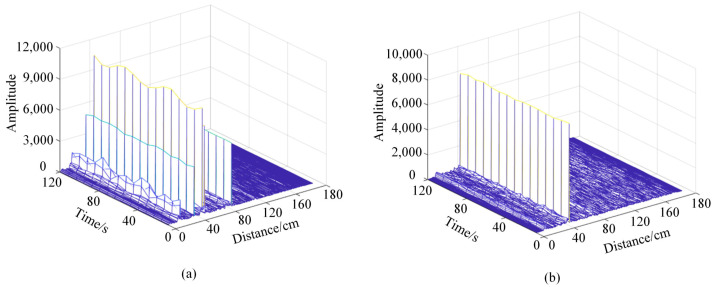
Radar echo signals before and after MTI processing static clutter: (**a**) Raw echo signal of millimeter wave radar. (**b**) The echo signal after MTI processing.

**Figure 3 sensors-23-00338-f003:**
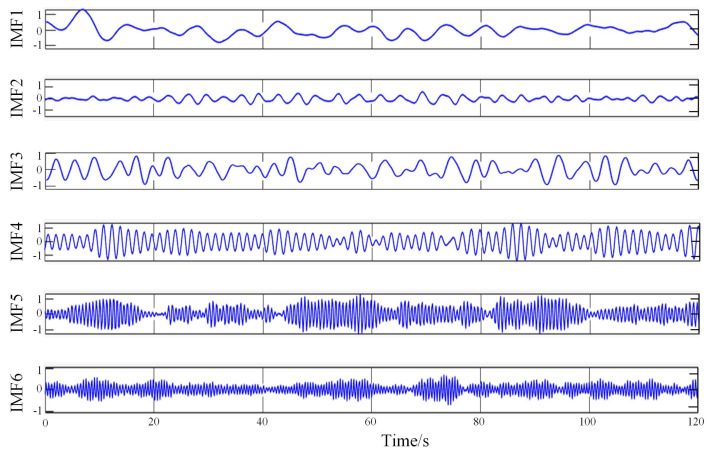
A set of results obtained by the VMD algorithm.

**Figure 4 sensors-23-00338-f004:**
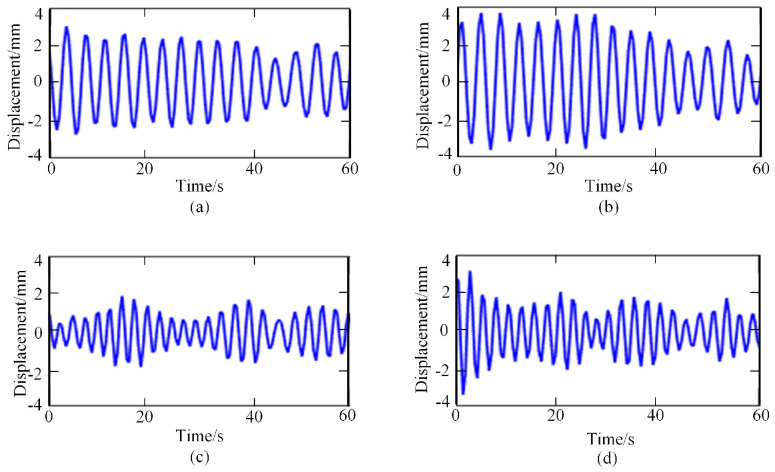
Respiratory signals under different emotional states: (**a**) relaxed; (**b**) happy; (**c**) sad; (**d**) anger.

**Figure 5 sensors-23-00338-f005:**
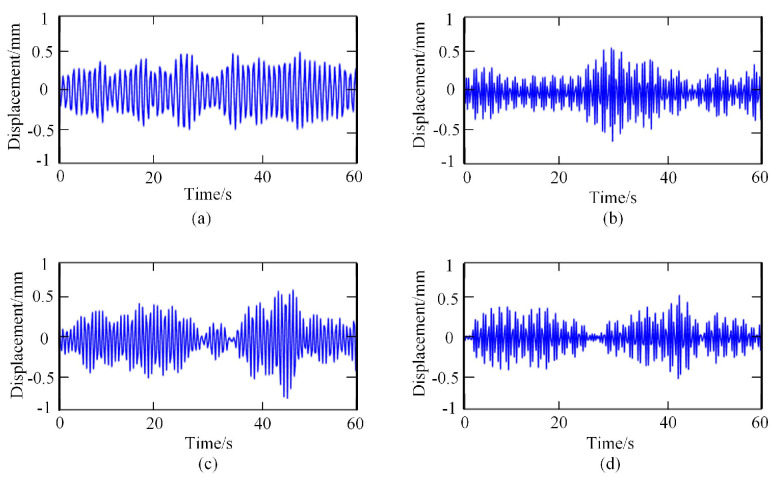
Heartbeat signals under different emotional states: (**a**) relaxed; (**b**) happy; (**c**) sad; (**d**) anger.

**Figure 6 sensors-23-00338-f006:**
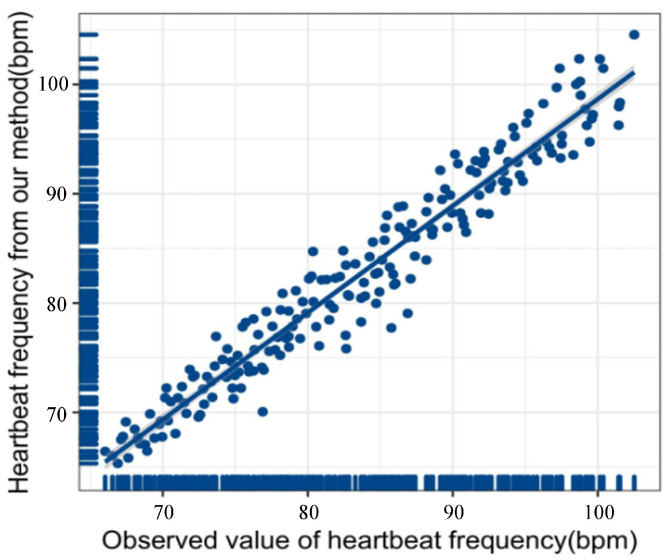
Linear regression image of heart rate calculated according to the method in this paper and heart rate monitored by Mi5 smart watch.

**Figure 7 sensors-23-00338-f007:**
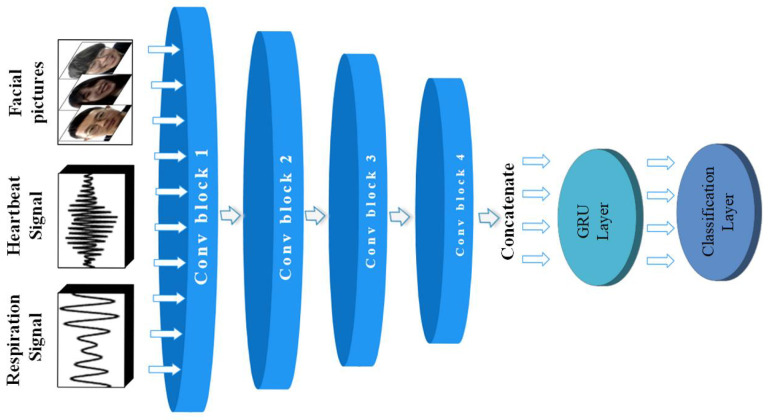
The overall structure of ER-MiCG deep learning model.

**Figure 8 sensors-23-00338-f008:**
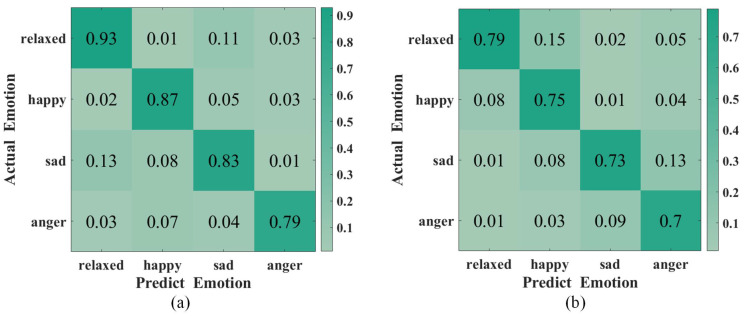
Confusion matrix for two different types of classifiers: (**a**) Person-dependent. (**b**) Person-independent.

**Figure 9 sensors-23-00338-f009:**
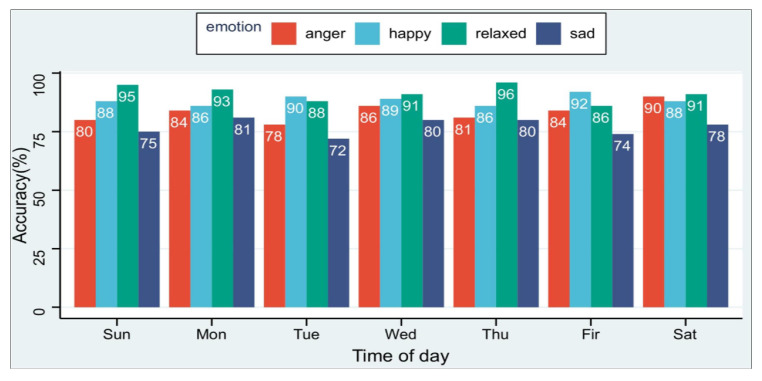
ER-MiCG’s classification performance of four emotion recognition in seven days a week.

**Figure 10 sensors-23-00338-f010:**
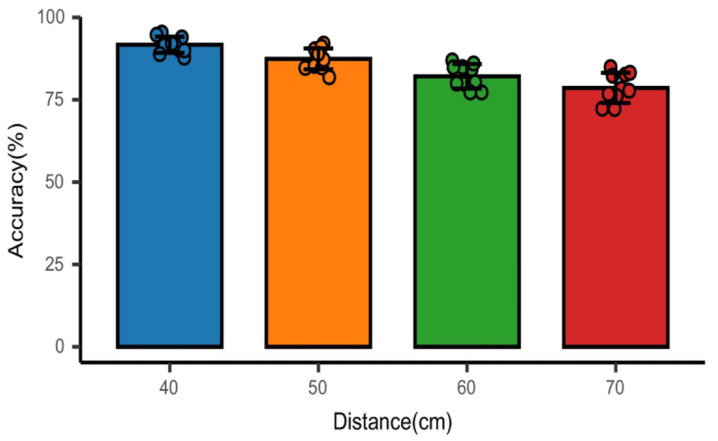
Effect of millimeter wave radar and subject on recognition accuracy at different distances.

**Figure 11 sensors-23-00338-f011:**
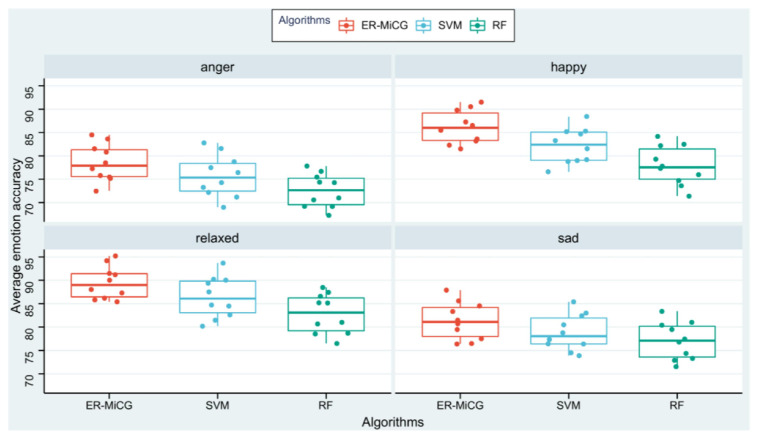
Comparison of ER-MiCG with two machine learning algorithms.

**Figure 12 sensors-23-00338-f012:**
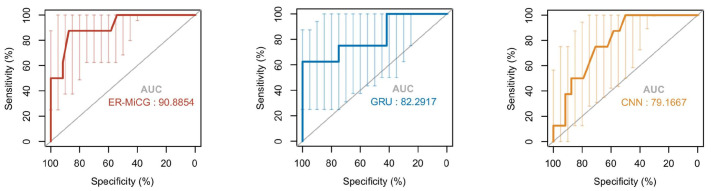
ROC curve of ER-MiCG with CNN and GRU.

**Table 1 sensors-23-00338-t001:** Network parameter settings for 1D-CNN.

Types	Kernel Size	No. of Filters	Stride
1D Full convolution	5	16	1
Max pooling	4	16	4
1D Full convolution	5	32	1
Max pooling	2	32	2
1D Full convolution	5	32	2
Max pooling	2	32	2
1D Full convolution	5	64	2
Max pooling	2	64	2

**Table 2 sensors-23-00338-t002:** Network parameter settings for 2D-CNN.

Types	Kernel Size	No. of Filters	Stride
2D Full convolution	3 × 3	32	1
Max pooling	2 × 2	32	2
2D Full convolution	3 × 3	64	1
Max pooling	2 × 2	64	2
2D Full convolution	3 × 3	96	1
Max pooling	4 × 4	96	4
2D Full convolution	3 × 3	96	1
Max pooling	4 × 4	96	4

**Table 3 sensors-23-00338-t003:** Radar setting parameters.

Parameters	Kernel Size
Number of Transmitting Antennas	1
Number of Receiving Antennas	4
Carrier Frequency	77 GHz
Bandwidth	4 GHz
frequency modulation	66.62 MHz/μs
Single Chirp Signal Duration	60 ηs
Period of frame	50 ms
Number of Chirps per Frame	128
Number of Frames	150
Number of Samples per Chirp	256

**Table 4 sensors-23-00338-t004:** Comparison of recognition accuracy of each model.

Project	Emotoion	Algorithm	Feature	Average Accuracy
ER-MICG	Relax, Happy, Sad, Anger	CNN and GRU	FMCW and FER	74.25%
EQ-Radio [43]	Joy, Pleasure, Sad, Anger	SVM	FMCW	72.3%
EmoSense [44]	Happy, Sad, Anger, Fear	KNN	CSI	40.86%
Yang Hao et al. [45]	Scary, Relax, Joy, Disgust	CNN and LSTM	FMCW and Continuous wavelet images	71.67%
Jain, N., et al. [46]	Fear, Happy, Sad, Anger, Surprise, Disgust, Neutral	CNN and RNN	FER	93.49%

## Data Availability

Not applicable.

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
