# Peer review of "Wireless Sensing Technology Combined with Facial Expression to Realize Multimodal Emotion Recognition"

_sensors, 2022, doi:10.3390/s23010338_

Round 1
Reviewer 1 Report
In my opinion, the paper is well written and has good technical components and clearly described but a rewrite is required before accept. I have some suggestions and questions.
Comment #1: A mathematical proof is needed how do the proposed technique outperform the existing models?
Comment #2: In the discussion, please comment on putting your system into the clinical routine. Is this planned - what steps are missing? Or has it happened already - How the physicians are using it, and what is their opinion on the tool?
Comment #3: Mention the limitations and future works of the developed system elaborately.
Comment #4: Please clarify the multimodal term. What do you mean by it and what's your contribution regarding it? How do the fusion happen?
Comment #5: The following references must be cited in Introduction section to describe the deep learning based systems for EEG data.
Islam, Md Rabiul, et al. "EEG Channel Correlation Based Model for Emotion Recognition." Computers in Biology and Medicine 136 (2021): 104757.
Islam, Md Rabiul, et al. "Emotion recognition from EEG signal focusing on deep learning and shallow learning techniques." IEEE Access 9 (2021): 94601-94624.
Comment #6: Methodology is not clear. Provide an algorithm and flowchart of the whole work. The authors need to add a new figure to show the main structure of the proposed system. This will help the reader to get a better understanding of what is going on in the proposed system.
Comment #7: Comparison with the previous works are highly required.
Comment #8: Abstract is unnecessarily wordy. Make it brief and concise. Also, Conclusion should clearly state the outcome. Some of the obtained results need to be highlighted in the conclusion section.
Comment #9: There are lots of typos. English needs to revise again with a professional editing service. Also, the figures are not clear in some cases.
Author Response
Dear expert teacher, Hello!
Thank you very much for your care of our research group, and also for your very strict and professional review and guidance of manuscript SENSORS-2057490.
The revision work of manuscript SENSORS-2057490 has been completed. The manuscript has been revised in detail according to the revision opinions of manuscript reviewers and editor teachers. In the editable revision, changes are marked accordingly. If you find any problems, please contact us in time.
I wish you good health, smooth work and a happy family!
Kind regards,
All authors of manuscript SENSORS-2057490.

Reviewer 2 Report
Wireless sensing technology combined with facial expression to realize multimodal emotion recognition
In this study the authors propose combining heartbeats, breading patterns, and facial expressions to recognize people’s emotions. According to the study results, the proposed method offers better results than unmixed conventional methods.
General comments
I like the way the study is justified in the introduction; especially regarding the modification of life patterns due to the COVID-19 pandemic. The existing technology is properly described and referenced. In general, I don not find big issues in this manuscript.
In the Experiments and results, there is the following paragraph: “A total of 900 groups of one-dimensional sequence information on heartbeat and respiration and related facial videos were collected in the experiment. Am ong them, 4 % of the data were abandoned due to blurred image quality and large movements of the subjects. The remaining 10 % of the data was separated as a validation set, 10 % as a test set and the other 76 % of the data was used as the training set.”
The numbers presented do not fit or are not properly presented. Nothing adds up 100% there! This is an important segment of the manuscript thus please present meaningful and readable numbers.
In the Conclusions the statement “This will undoubtedly promote the development of wireless sensor technology in emotion recognition and will also profoundly impact the development of human-computer interaction ( HCI ), modern health care and education.” Is somewhat speculative. Consider changing the phrase to a softer expectation of the future.
Observations about writing
Should the authors add punctuation to the equation, the readability would improve. Equation punctuation refers to adding a period or a comma after each equation. This may affect the text coming after the equation; for example, if you use a comma after equation (3), then (line 164) would start with lowercase ( ‘where’ with no bleeding instead of ‘Where’).
I would not use acronyms in the Keywords.
I would not use acronyms in the Abstract.
Line 77: “The deep learning model combined with CNN…”. Most references indicate Deep Learning consist of using Artificial Neural Nets to adjust the machine performance to the modeled system. Therefore, what could be the actual meaning of combining Deep Learning with CNN?
Not all the acronyms are included in the Abbreviations list.
The phrase “With more and more public data sets providing various emotions-related” appears in more than one instance. Please vary the writing.
Punctuation mistake in line 392. After Figure 10.
Final remarks
The manuscript is well-written and organized. Despite presenting many places where the manuscript can be improved, all these problems are minor and do not affect the actual content of the paper. In my opinion, this manuscript will make merit for publication after the minor writing problems are corrected.
Author Response

(The authors gave the same response as above.)

Reviewer 3 Report
The paper “Wireless sensing technology combined with facial expression to realize multimodal emotion recognition” combined the data from facial expression images with heartbeat and breathing signals collected by millimetre-wave radar in order to obtain a CNN+GRU model able to detect four facial expressions.
However, most of the facial expression recognition research focus on detecting at least six basic emotions: neutral, happy, sad, angry, surprise, fear. In addition, several models can also detect disgust, contempt or other states. The author should What was the reason for choosing the four states in the paper(relaxed, happy, anger, sad).
The accuracy of 84.5% bellow state of the art emotion recognition models which can detect at least 6 states. For example, deep learning FER models can have accuracies over 90%. A comparison with state of the art researches in FER should be carried out.
In addition there are some misspellings that should be fixed, e.g.:
· Line 449- indicator instead of “indicator”
· Sections 2.2.1, 2.2.2 and 2.2.3. have the same name, “MTI removes static clutter”
· In Fig.4 and 5 happy, sad and anger are denoted with the same letter(“b”) in the list
Author Response

(The authors gave the same response as above.)

Round 2
Reviewer 1 Report
Well revised.
Reviewer 3 Report
The paper has been revised in detail according to the observations. Thus, the paper can be published in the journal